# Competition between Variable–Supply and Fixed–Supply Currencies

Guizhou Wang and Kjell Hausken *

Faculty of Science and Technology, University of Stavanger, 4036 Stavanger, Norway
* Correspondence: kjell.hausken@uis.no

**Abstract:** For one variable–supply currency in isolation, one player's Cobb–Douglas utility depends on the current supply divided by the initial supply, multiplied by the inverse of the accumulative inflation/deflation. With equal weight assigned to both factors, money printing outweighs inflation, and money withdrawal outweighs deflation. The study design is to analyze how competition between one variable–supply and one fixed–supply currency impacts the player's choice of currency. Applying the 1959–2021 US M2 money supply data and the 1635–2021 US inflation data, the player's utility increases over time when assigning high weight to money printing/withdrawal and increases less or decreases overall when assigning high weight to inflation/deflation. With different player support for the two currencies, depending on each currency's backing, convenience, confidentiality, transaction efficiency, financial stability, and security, replicator dynamics is used to determine the player's volume fraction of transactions in each currency. Low, high, increasing, and decreasing support of a currency are analyzed. Each fraction may increase, decrease, be inverse U–shaped, U–shaped, and approach low or high levels over time. For example, high weight assigned to money printing may cause the player to eventually prefer the variable–supply currency unless the player supports the fixed–supply currency highly and increasingly.

**Keywords:** digital currencies; currency competition; money supply; inflation; replicator dynamics; cryptocurrencies; central bank digital currencies





## 1. Introduction

### 1.1. Background

The emergence of new digital currencies raises questions about how these will compete depending on their characteristics. Historically, currencies have been associated with nations, such as the USD, CNY, EUR, etc. Nakamoto (2008) demonstrated successfully how a decentralized currency (Bitcoin) can be successfully built on a blockchain by applying proof of work technology with no centralized authority. Thereafter 20,178 cryptocurrencies have emerged (with a market cap of USD 915 billion) with great variation in the degree of decentralization, consensus mechanisms (e.g., proof of stake), supply, burning of coins, etc.[1] The introduction of such currencies, combined with central banks expanding their digital currencies, changes the nature of currency competition. Currencies can have all kinds of characteristics related to supply, ownership, decentralization, regulation, confirmation of transactions, geographical extension, etc.

The Federal Reserve Bank of Boston has conducted payment surveys since 2008. According to the latest 2020 Survey of Consumer Payment Choice (Foster et al. 2021), in 2020, consumers in the US, on average, made 68 payments per month. The top three payment methods are debit cards (23 payments) and credit cards (18 payments), followed by cash (14 payments). These three payment methods account for 80% of all payments by numbers (Greene and Stavins 2021). In 2020, cash accounts for 19% of all payments, a drop of 7% from 2019. This illustrates how payment methods can evolve within fiat currencies.

*1.2. Contribution*

This article analyzes currency competition focusing explicitly on supply and inflation. One variable–supply currency is considered where money can be printed and withdrawn and be subject to inflation or deflation. Both these two concerns have historically been important. Money printing enabled by a variable–supply currency offers additional options not available for a fixed–supply currency. One example is Roosewelt's 1933–1939 New Deal to recover the economy. Another example is war funding, e.g., World War I and World War II. The additional options may cause disadvantages. For example, USD 1 in 2022 buys 1.22% of what it would buy in 1695, which is a poor store of value for this time period. Variable–supply currencies have historically not implemented mechanics to ensure that one unit of a currency generates the same purchasing power on average over certain periods of time. Theoretically, such mechanics would enable financing a New Deal or a war with money printing if corresponding money withdrawals were implemented thereafter. A variable–supply currency with such mechanics would be a better store of value.

As a benchmark competitor, a fixed–supply currency is considered where money printing/withdrawal and inflation/deflation are impossible. Such a currency may be a good store of value and may potentially compete with a variable–supply currency which may lose its purchasing power over time. Historically, a fixed–supply currency has been close to impossible. The closest has been gold, which scores higher than Bitcoin on established history, and scores lower than Bitcoin on portability, divisibility, censorship resistance, verifiability, and scarcity (Ikkurty 2019). Both gold and Bitcoin score high on durability and fungibility (BYBIT Learn 2021). Gold, which is a currency under the current system (Mitchell 2021), has historically approximated fixed supply, with 1.5% additional gold mined in 2020.[2] Bitcoin has a fixed supply of 21 million coins. As of January 2022, 18.9 million Bitcoin have been mined, i.e., 90% (Hayes 2022). The remaining 2.1 million Bitcoin will be mined until approximately 2140.

First, a variable–supply currency is analyzed in isolation. A player's Cobb–Douglas utility is a product of two ratios. The first ratio is the initial supply plus accumulative money printing/withdrawal in the numerator, divided by the initial supply. The second ratio equals the inverse of accumulative inflation/deflation. With equal weight to both ratios, a utility of 1 constitutes a benchmark that is exceeded by assigning more weight to money printing, which can be useful to recover or boost the economy. The utility is less than 1 when assigning more weight to inflation, which is useful when seeking to cool down the economy. The US M2 money supply since 1959 and US inflation since 1635 are used to show how a player's utility increases or decreases over time depending on the player's preferences.

Thereafter competition between a variable–supply currency and a fixed–supply currency is analyzed. The two currencies may have different support depending on their backing, convenience, confidentiality, transaction efficiency, financial stability, and security. Replicator dynamics is used to analyze how the player's fractions of transacting in each currency evolve over time depending on the weights assigned to money printing and inflation and whether the support for each currency is constant, increases or decreases over time. Such insight is useful for policy makers and others seeking to determine how to adjust money printing, inflation, and support for various currencies.

## 2. Literature

The limited literature on this topic is divided into five groups, i.e., currency competition, competition between fiat currencies and cryptocurrencies, CBDCs (central bank digital currencies) and cryptocurrencies, the cryptocurrency market, and game theoretic analyses and decision models.

*2.1. Currency Competition*

Dowd and Greenaway (1993) develop a framework to analyze currency competition focused on network effects and switching costs. They find that network effects and switch-

ing costs seem to make it optimal for an agent to adopt only one currency. The agent is often reluctant to abandon the existing currency even if it is manifestly inferior to a new currency. They argue that parallel currencies are relatively uncommon. Camera et al. (2004) explore the competition between one safe foreign fiat currency, such as the US dollar, and one risky home fiat currency in a decentralized trading environment. They find that traders normally prefer safe foreign currency unless the trade frictions are high. A risky home currency in a poorly functioning economy is prone to dollarization. Dollarization can be reduced by adopting policies aimed at reducing currency risk and enhancing the trading environment. Gawthorpe (2017) adopts the money in utility function approach to explore the competition between a fiat currency and alternative currencies. They show that competition may cause a lower inflation rate compared with only one fiat currency. Wang and Hausken (2021) investigate the competition between a national currency and a global currency among three types of players, i.e., conventionalists, pioneers, and criminals. They consider six utility features of a currency, i.e., backing, convenience, confidentiality, transaction efficiency, financial stability, and security. They also apply replicator dynamics to analyze the evolution of the fractions of the three kinds of players and how they choose among the two currencies.

This article contributes to this literature by considering the competition between a variable–supply currency and a fixed–supply currency. The article focuses mainly on two variable–supply currency features, i.e., money printing/withdrawal and inflation/deflation. Other features, such as backing, convenience, safety, privacy, etc., are also implicitly embedded in the model.

### 2.2. Competition between Fiat Currencies and Cryptocurrencies

Wang and Hausken (2022b) analyze the evolution of fixed–supply and variable–supply currencies. The latter enable money printing/withdrawal and inflation/deflation. They find that a player's utility of transacting in each currency is proportional to how the player supports that currency, the volume fraction of all the players' transactions in that currency, and the fraction of players of the same kind as the given player. The current article contributes three advances over Wang and Hausken (2022b). First, if inflation empirics are unavailable, we estimate inflation from money printing by assuming a time lag. Second, if money printing empirics are unavailable, we estimate money printing from inflation by assuming a time lag in the opposite time direction. Third, this article purifies the analysis of how one kind of player supports one currency relative to the other currency, while Wang and Hausken (2022b) consider how two kinds of players support one currency relative to the other currency differently. The analysis of one kind of player enables focusing explicitly on how one typical or average player reacts to money printing/withdrawal and inflation/deflation depending on supporting the two currencies equivalently or differently.

Schilling and Uhlig (2019) analyze agents choosing between a fiat currency and a cryptocurrency. For example, fiat currencies are currently useful for most purchases, while cryptocurrencies may enable tax evasion, anonymity, and censorship resistance. Value–added tax and transaction fees to miners also play a role. They find that substitution decreases with asymmetry in exchange fees and transaction costs. Their analysis corresponds to the different support for the two currencies analyzed in this article, which depends on the currencies' transaction efficiencies.

Fernández-Villaverde and Sanches (2019) consider competition between privately issued fiat currencies. They determine a price stable equilibrium for multiple currencies in a Lagos–Wright environment, corresponding to two coexisting currencies in the current article and various less desirable equilibria. Almosova (2018) supplements their model by assuming costly circulation of private currencies due to mining costs, verification of transactions, etc. Although cryptocurrency competition will not cause price stability, with less costly private currency circulation, competition will cause downward pressure on the inflation of the public currency. Rahman (2018) investigates how fiat and digital currency

competition impact monetary policy. He finds that a socially efficient allocation cannot follow from a purely private arrangement of digital currencies.

Lagos and Wright (2005) propose a framework for policy analysis based on the frictions that are essential for money. They allow the agents to interact periodically in both decentralized and centralized markets. Their model estimates that the welfare cost of inflation equals 3–5% of consumption. The framework can be used to analyze how the different regimes, such as one currency versus two currencies, cause different outcomes. Benigno et al. (2022) analyze two national currencies and a global cryptocurrency. They find that deviating from interest rate equality may imply approaching the zero lower bound or abandoning the national currency. They conclude that simultaneously ensuring a fixed exchange rate, free capital flows, and an independent monetary policy becomes even less possible. Verdier (2021) analyzes how a digital currency impacts competition in the deposit and lending markets. She finds increasing bank lending rates as a consequence of the digital currency crowding out bank deposits.

Hong et al. (2018) investigate the potential crowding out effect in a regime consisting of a fiat currency and a digital currency. The crowding–out effect occurs only under extreme conditions, i.e., high costs for one currency and low costs for the other currency. Obu and Ukpere (2022) investigate the impact of cryptocurrencies on the effectiveness of the fiscal policy. They find that government purchases decrease with households' adoption of cryptocurrencies. Sissoko (2021) explores the conceptual world where currencies are convertible into the numeraire consumption goods at a fixed rate. Then nobody wants to hold money over time. He points out that it is possible to establish a banking system in such an environment. The ability to increase the money supply according to societal needs is essential for the banking system's efficiency.

This article contributes to this literature by considering how a variable–supply fiat currency competes with a fixed–supply currency such as Bitcoin. Changing supply and inflation/deflation for the variable–supply fiat currency is explored, together with how the player chooses between a variable–supply currency and a fixed–supply currency over time. The replicator equation is applied to show the dynamic evolution of the volume fractions of the two currencies.

### 2.3. CBDCs and Cryptocurrencies

Caginalp and Caginalp (2019) analyze asset allocation between a home currency and a cryptocurrency when the government confiscates some of the players' assets. Blakstad and Allen (2018) evaluate which possibilities and risks cryptocurrencies offer for central banks and individuals and the challenges of issuing CBDCs. Masciandaro (2018) assess how different media of payments may evolve depending on individual preferences, akin to how the two currencies in the current article may evolve over time. Belke and Beretta (2020) suggest that central banks need to embrace the technology underlying cryptocurrencies. They suggest that central banks issuing cryptocurrencies may be subject to the disadvantages of cryptocurrencies and few benefits. Benigno (2021) reasons that currency competition causes the nominal interest rate and inflation to be determined by the time discount factor, the exit rate, and the fixed cost of entry, which can challenge the function of central banking. Asimakopoulos et al. (2019) find a substitution effect between the real balances of government currency and cryptocurrencies as a consequence of preferences, technology, and monetary policy shocks. This article relates to this literature by assessing CBDCs and cryptocurrencies from the supply perspective. A CBDC is usually a variable–supply currency. A cryptocurrency such as Bitcoin is a fixed–supply currency. The article presents a model that shows the competition and evolution of a variable–supply currency and a fixed–supply currency.

### 2.4. The Cryptocurrency Market

ElBahrawy et al. (2017) evaluate the fluctuating evolution of market shares of 1469 cryptocurrencies between April 2013 and May 2017. Caporale et al. (2018) determine a

positive correlation between cryptocurrencies' past and future values. ElBahrawy et al. (2019) assess the linkage between online attention towards digital currencies on Wikipedia and market dynamics for digital currencies. White (2014) assesses the different market shares of Bitcoin and altcoins, akin to the current article assessing the volume fractions of transactions for two currencies. Sapkota and Grobys (2021) find no relation between the submarket equilibria of privacy coins and non–privacy coins for the top ten cryptocurrencies in 2016–2018. Milunovich (2018) estimates weak connectedness between five popular cryptocurrencies as one group and six major asset classes as a second group, and strong connectedness within each group, with a few exceptions. Gandal and Halaburda (2016) determine no winner–take–all effects in early cryptocurrency competition and strong network effects and winner–take–all dynamics more recently. The best well–known cryptocurrency is Bitcoin. It has a limited supply of 21 million coins. This article relates to this literature by assessing the competition between a variable–supply currency, such as fiat money, and a fixed–supply cryptocurrency, such as Bitcoin. The market share of the two currencies is captured by the volume fractions of the two currencies. The model shows the dynamic evolution of the market shares of the two currencies.

### 2.5. Game Theoretic Analyses and Decision Models

Imhof and Nowak (2006) analyze a stochastic frequency–dependent Wright–Fisher process to specify which of two strategies survive. They find that the Markov process has two absorbing states corresponding to homogeneous populations choosing either strategy A or strategy B. Lewenberg et al. (2015) find that it is difficult or impossible to distribute rewards in a stable way for a pooled Bitcoin mining and rewards cooperative game and that players continuously prefer to switch pools. Wang and Hausken (2022a) present a two–period decision model between a central bank and a household. They analyze the household's asset portfolio choice among production, consumption, CBDC, and non–CBDC, such as Bitcoin. This article related to this literature by considering a player's choice between a variable–supply currency and a fixed–supply currency and how this choice is made over time.

### 2.6. Literature Summary and Additions to the Literature Gap

The literature commonly analyzes the competition between currencies and focuses on different currencies' features, i.e., network effects and switching costs (Dowd and Greenaway 1993), safety, risk, and trade frictions (Camera et al. 2004), switching costs, inflation and network externalities (Gawthorpe 2017), six utility features of a national currency and a global currency (Wang and Hausken 2021), etc. This is one of the first articles that focuses on two essential features of currencies, which are supply and inflation/deflation. Thus, this article adds to this literature gap by exploring currency competition from the supply and inflation/deflation perspective.

Recent literature explores the competition between fiat currencies and cryptocurrencies, e.g., the substitution effects under asymmetry in transaction costs (Asimakopoulos et al. 2019; Schilling and Uhlig 2019), the coexistence and equilibrium of multiple currencies (Fernández-Villaverde and Sanches 2019), the impact on monetary policy and fiscal policy (Benigno et al. 2022; Obu and Ukpere 2022; Rahman 2018), the impact on the deposit and lending market (Verdier 2021), and the crowding out effects under a multiple currencies regime (Bian et al. 2021; Hong et al. 2018). In addition, the literature commonly investigates the relationship between CBDCs and cryptocurrencies (Belke and Beretta 2020; Benigno 2021; Blakstad and Allen 2018). The existing literature barely explores the player's choice between two currencies with respect to the supply and inflation/deflation features. This article adds to this literature gap by demonstrating the evolution of a player's choice between a variable–supply currency and a fixed–supply currency over time. The analysis mainly focuses on the supply and inflation/deflation features and incorporates how the player supports one currency relative to the other currency.

The literature furthermore evaluates the cryptocurrency market, e.g., the market shares of Bitcoin and altcoins (White 2014), the evolution of cryptocurrencies' market shares (ElBahrawy et al. 2017), and the equilibria of the cryptocurrency market (Sapkota and Grobys 2021; Yi et al. 2022). This article fills this literature gap by investigating how the market share of a fixed–supply cryptocurrency such as Bitcoin evolves over time in competition with a variable–supply currency. The market share is represented by the currency's transaction volume. Game theoretic models and decision models are widely used in academic research (Hausken and Welburn 2022; Imhof and Nowak 2006; Prat and Walter 2021; Wang and Hausken 2022a). This article adds to this literature by demonstrating a player's choice of a variable–supply currency versus a fixed–supply currency and the dynamic evolution of the volume fractions of the two currencies over time.

## 3. The Model

The article models one player receiving different Cobb–Douglas utilities depending on its choice of either a variable–supply fiat currency or a fixed–supply currency. The player mainly considers the two features of a currency, i.e., printing/withdrawal and inflation/deflation. Additional factors, i.e., transaction efficiency, banking, anonymity, security, confidentiality, finality, and stability, are comprised of one parameter, which expresses the player's support of a variable–supply currency relative to a fixed–supply currency.

The six dependent or outcome variables are the player's Cobb–Douglas utility of holding a fixed–supply currency, the player's Cobb–Douglas utility of holding a fixed–supply currency when the variable–supply currency is subject to money printing, the player's Cobb–Douglas utility of holding a fixed–supply currency when the variable–supply currency is subject to inflation, the player's Cobb–Douglas utility of holding a fixed–supply currency when a variable–supply currency is available, the player's Cobb–Douglas utility of holding a variable–supply currency, and the player's Cobb–Douglas utility of holding both a variable–supply currency and a fixed–supply currency in a certain weighted combination. The dynamic competition between a fixed–supply currency and a variable–supply currency is presented by the evolution of the volume fraction of the player's transactions in the variable–supply fiat currency using the replicator equation. The model demonstrates how a variable–supply currency competes with a fixed–supply currency over time.

### 3.1. One Variable–Supply Fiat Currency n

Consider a fiat currency, which may be a national currency with variable supply $s_i$ at the discrete times $i = t_0, t_0 + 1, t_0 + 2, \ldots, T$, where $t_0 \geq 0$ and any time interval of length 1 applies, e.g., year, month, week, day, etc., and $T$ is the final time. Hence $s_{i+1} - s_i$ is the amount printed (if positive) or withdrawn (if negative) from time $i$ to time $i + 1$. Summing up, $\sum_{i=t_0}^{t-1}(s_{i+1} - s_i)$ is the amount printed or withdrawn from time $i = t_0$ to time $i = t - 1$. Hence $\frac{s_{t_0} + \sum_{i=t_0}^{t-1}(s_{i+1} - s_i)}{s_{t_0}}$ is the money supply at time $t$ divided by the money supply at time $t_0$, which can be considered as a player's purchasing power at time $t$ relative to the purchasing power at time $t_0$ without inflation.

Assume inflation $\pi_i$ at time $i = t_0, t_0 + 1, \ldots, T$. Hence an asset valued at 1 at time $i = t_0$ is valued as $\frac{1}{1 + \pi_{t_0+1}}$ at time $i = t_0 + 1$, $\frac{1}{(1 + \pi_{t_0+1})(1 + \pi_{t_0+2})}$ at time $i = t_0 + 2, \ldots$, and $\frac{1}{\prod_{i=t_0+1}^{t}(1 + \pi_i)}$ at time $i = t$, which is the degraded asset value due to accumulative inflation from time $t_0$ to time $t$. The terms $\frac{s_{t_0} + \sum_{i=t_0}^{t-1}(s_{i+1} - s_i)}{s_{t_0}}$ and $\frac{1}{\prod_{i=t_0+1}^{t}(1 + \pi_i)}$ are not stationary. Instead, they are affected by the currency supply $s_t$ and the inflation $\pi_t$. Thus, both terms evolve over time.

Multiplying $\frac{s_{t_0} + \sum_{i=t_0}^{t-1}(s_{i+1}-s_i)}{s_{t_0}}$ raised to the Cobb–Douglas elasticity $\alpha$, $0 \leq \alpha \leq 1$, with the degraded asset value $\frac{1}{\prod_{i=t_0+1}^{t}(1+\pi_i)}$ raised to the Cobb–Douglas elasticity $1 - \alpha$ gives the player's Cobb–Douglas utility

$$u_{nt} = \left( \frac{s_{t_0} + \sum_{i=t_0}^{t-1}(s_{i+1}-s_i)}{s_{t_0}} \right)^\alpha \left( \frac{1}{\prod_{i=t_0+1}^{t}(1+\pi_i)} \right)^{1-\alpha} \tag{1}$$

at time $t$ for holding a fiat currency $n$ subject to variable money supply $s_t$ and inflation $\pi_t$ at time $t$, $t \geq t_0$. If $\alpha > 0.5$, the player assigns more weight to advantageous purchasing power than to disadvantageous inflation, and conversely if $\alpha < 0.5$. The player weighs the two considerations against each other. Equal weights $\alpha = 0.5$ is an especially interesting benchmark since the constant utility $u_{nt} = 1$ can be envisioned where the player's increased purchasing power from money printing $s_{i+1} - s_i$ is exactly offset by inflation $\pi_i$ through time, or money withdrawal $s_{i+1} - s_i$ is exactly offset by deflation $\pi_i$ through time. If inflation $\pi_t$ is strictly positive in the long run (i.e., $\pi_\infty > 0$), then the utility $u_{nt}$ converges to zero, i.e., $\lim_{t \to \infty} u_{nt} = 0$. This property holds only when inflation $\pi_t$ is sufficiently high through time $t$, i.e., when the impact of inflation $\pi_t$ is greater than the impact of the currency supply $s_t$. In the long run, the evolution of the utility $u_{nt}$ depends on the currency supply $s_t$ and the inflation $\pi_t$.

Overall, (1) expresses the player's Cobb–Douglas utility from the currency supply $s_t$ and inflation $\pi_t$. This conception captures reality to some extent. For the player, a higher inflation $\pi_t$ means currency devaluation. Thus, the player's utility $u_{nt}$ decreases with inflation $\pi_t$. This article adopts the money–in–the–utility approach as in (1). It is one of the fundamental approaches in academic research, especially in economics and finance. The money–in–the–utility approach has a long history and is an important tool in economic research. The idea is that the utility function measures the player's preferences on a basket of goods and services. As an early pioneer, Ramsey (1928) assumes that the representative agent makes decisions by maximizing its utility. Sidrauski (1967) similarly conceptualizes a money–in–the–utility function. More recent examples are Block and Heineke (1975); Chen and Guo (2014); Ganelli and Tervala (2010); Mian et al. (2021); Obstfeld (1981); Wachter and Yogo (2010).

If inflation empirics are unavailable, and money printing empirics prior to time $t_0$ are unavailable or ignored, inflation can be estimated from money printing. Assume that money printing at time $i$ gives inflation at time $i + \tau$, $\tau \geq 0$. Hence, when $t - t_0 > \tau$, we invert the ratio for the player's purchasing power at time $t$ relative to the purchasing power at time $t_0$ without inflation, and account for the time delay of $\tau$ by summing from $i = t_0 + \tau$ to $i = t - 1$, instead of summing from $i = t_0$ to $i = t - 1$. Hence, no inflation occurs from time $t_0$ to time $t_0 + \tau$. Equation (1) is thus replaced by

$$u_{nMt} = \begin{cases} \left( \frac{s_{t_0} + \sum_{i=t_0}^{t-1}(s_{i+1}-s_i)}{s_{t_0}} \right)^\alpha & if \ t - t_0 \leq \tau \\ \left( \frac{s_{t_0} + \sum_{i=t_0}^{t-1}(s_{i+1}-s_i)}{s_{t_0}} \right)^\alpha \left( \frac{s_{t_0}}{s_{t_0} + \sum_{i=t_0+\tau}^{t-1}(s_{i+1}-s_i)} \right)^{1-\alpha} & if \ t - t_0 > \tau \end{cases} \tag{2}$$

where, evidently, the inflation term vanishes when $t - t_0 \leq \tau$.

If money printing empirics are unavailable, and inflation empirics prior to time $t_0$ are unavailable or ignored, money printing can be estimated from inflation. Assume that inflation at time $i + \tau$ is due to money printing at time $i$. For the inflation term, we sum

from $i = t_0 + 1 + \tau$ to $i = t$ instead of summing from $i = t_0 + 1$ to $i = t$. Hence, no inflation occurs from time $t_0$ to time $t_0 + \tau$. Equation (1) is thus replaced by

$$u_{nIt} = \begin{cases} \left( \prod_{i=t_0+1}^{t} (1 + \pi_i) \right)^{\alpha} & if \ t - t_0 \leq \tau \\ \left( \prod_{i=t_0+1}^{t} (1 + \pi_i) \right)^{\alpha} \left( \frac{1}{\prod_{i=t_0+1+\tau}^{t}(1+\pi_i)} \right)^{1-\alpha} & if \ t - t_0 > \tau \end{cases} \quad (3)$$

### 3.2. One Variable–Supply Fiat Currency n Competing with One Fixed–Supply Currency g

Assume that a variable–supply fiat currency $n$ competes with a fixed–supply currency $g$, which may be a global currency, e.g., Bitcoin, which eventually (in ca. year 2140) has a fixed supply of 21 million coins. A player comparing which of two currencies to use will account for additional factors beyond money printing and inflation. We comprise these factors into one parameter $h_t$, $0 \leq h_t \leq 1$, at time $t$, which expresses the player's support of the fixed–supply currency $g$ relative to the variable–supply currency $n$ at time $t$. The player supports currency $g$ more than currency $n$ when $0.5 < h_t \leq 1$, supports currency $n$ more than currency $g$ when $0 \leq h_t < 0.5$, supports exclusively currency $g$ when $h_t = 1$, supports the currencies equally much when $h_t = 0.5$, and supports exclusively currency $n$ when $h_t = 0$.[3] Multiplying $1 - h_t$ with (1) gives the player's utility

$$u_{ngt} = \left( \frac{s_{t_0} + \sum_{i=t_0}^{t-1} (s_{i+1} - s_i)}{s_{t_0}} \right)^{\alpha} \left( \frac{1}{\prod_{i=t_0+1}^{t}(1 + \pi_i)} \right)^{1-\alpha} (1 - h_t) \quad (4)$$

for transacting with the fiat currency $n$.

Conversely, since currency $g$ is not subject to money printing and inflation, the two first terms in (4) disappear. Hence, the player's utility for transacting with the fixed–supply currency $g$ is

$$u_{gnt} = h_t \quad (5)$$

Assume that the player at time $t$ chooses a volume fraction $p_{nt}$ of its transactions to be in the variable–supply fiat currency $n$, and the remaining volume fraction $1 - p_{nt}$ to be in the fixed–supply currency $g$. The player's utility at time $t$ is thus the weighted combination

$$u_t = p_{nt} u_{ngt} + (1 - p_{nt}) u_{gnt} \quad (6)$$

One interesting aspect of the money–in–the–utility approach arises when multiple currencies may potentially coexist simultaneously. This article incorporates two currencies, i.e., a variable–supply currency $n$ and a fixed–supply currency $g$, assigned different weights or probabilities $p_{nt}$ and $1 - p_{nt}$. Thus, (6) captures the player's weighted utility $u_t$, accounting for two currencies.

### 3.3. Replicator Dynamics

To determine the evolution of the fraction $p_{nt}$ of the player's transactions in the variable–supply fiat currency $n$, we apply the replicator equation (Taylor and Jonker 1978; Weibull 1997, p. 69)

$$\frac{\partial p_{nt}}{\partial t} = k p_{nt} (u_{ngt} - u_t) = k p_{nt} (1 - p_{nt}) (u_{ngt} - u_{gnt}) \quad (7)$$

where (6) has been inserted. In (7), $k > 0$ is the sensitivity or rapidity of change of the process. When $k$ is intermediate, the process is stable. The process changes rapidly when $k$ is high, and slowly when $k$ is low. The right–hand side of (7) is proportional to the difference $u_{ngt} - u_t$ between the player's utility of using the variable–supply fiat currency $n$ and the weighted combination of both utilities in (6), and also proportional to the difference $u_{ngt} - u_{gnt}$ between the player's utility of using the variable–supply fiat currency $n$ and

the utility of using the fixed–supply currency $g$. Hence, when the former exceeds the latter, the fraction $p_{nt}$ increases and conversely decreases when the former is lower than the latter. The right–hand side of (7) is also proportional to the product $p_{nt}(1 - p_{nt})$ of both fractions, which is inverse U–shaped with a maximum at $p_{nt} = 0.5$ and minima when $p_{nt} = 0$ and $p_{nt} = 1$. Hence, the fractions $p_{nt}$ and $1 - p_{nt}$ change most rapidly when they are equally large, which means that the player chooses equal volume fractions $p_{nt} = 1 - p_{nt} = 0.5$ for the two currencies. The evolution of the fraction $p_{nt}$ of the player's volume of transactions in currency $n$ at time $t$ depends on the Cobb–Douglas elasticity $\alpha$ and the currency support parameter $h_t$. In the long run, only one currency survives. Specifically, the process always evolves toward one or the other currency, eventually surviving exclusionarily, which may take some time, dependent on the initial conditions, the sensitivity parameter $k$, and the model parameters.

## 4. Analyzing the Model

### 4.1. The US 1635–2021

Figure 1a shows the US M2 money supply $s_i$ at time $i$, i.e., 1959–2021, interpreted as M2, which includes currency, and certain deposit and money market accounts, increasing from USD 289.8 billion in January 1959 referred to as time $t_0$ to USD 21,425.9 billion in November 2021 referred to as time $T$ (Federal Reserve 2022). Figure 1b shows the US inflation $\pi_i$ at time $i$, i.e., 1959–2021, with a maximum 13% in 1980 and a minimum of 0% in 2009 and 2015 (CPI Inflation Calculator 2022). Figure 1c,d, with different time scales, insert the empirics in Figure 1a,b into (1) and plot the player's utility $u_{nt}$ for the five Cobb–Douglas elasticities $\alpha = 0.6,\ 0.5,\ 0.4,\ 0.3,\ 0.2$. More weight $\alpha = 0.6$ to money printing than inflation causes $u_{nt}$ to increase overall. The intermediate elasticity $\alpha = 0.5$, discussed after (1), is especially interesting. Equal weights assigned to money printing and inflation causes the player's utility $u_{nt}$ to increase overall from 1959 to 2021. When $\alpha = 0.4$, i.e., less weight is assigned to advantageous money printing than to disadvantageous inflation, the player's utility remains above utility $u_{nt} = 1$ throughout, reaching minima of $u_{nt} = 1.01$ in 1981 and 1996. When $\alpha = 0.3$, i.e., even less weight assigned to money printing than to inflation, the player's utility is initially inverse U–shaped and crosses below $u_{nt} = 1$ in 1974, remaining below $u_{nt} = 1$ thereafter. When $\alpha = 0.2$, the player's utility is $u_{nt} = 1.00$ in 1959 and 1960 (rising briefly to $u_{nt} = 1.01$ halfway through 1959). Thereafter $u_{nt}$ is inverse U–shaped, reaches $u_{nt} = 1$ in 1967, increases briefly to $u_{nt} = 1.02$ through 1967, and finally crosses below $u_{nt} = 1$ in 1968, where it remains thereafter.

Figure 1e assumes the time lag $\tau = 2$ years from money printing to inflation and insert the money printing empirics in Figure 1a into (2) and plot the player's utility $u_{nt}$ for the five Cobb–Douglas elasticities $\alpha = 0.6,\ 0.5,\ 0.4,\ 0.3,\ 0.2$, thus not applying the inflation empirics. Batini (2006), Batini and Nelson (2001) and Friedman and Schwartz (1982) find that it takes more than one year from money printing until inflation. Figure 1e gives overall lower player utility than Figure 1c, possibly because inflation estimated from money printing may cause more estimated inflation than the empirical inflation in Figure 1b. The benchmark elasticity $\alpha = 0.5$, i.e., equal weights assigned to money printing and inflation, causes the player's utility $u_{nt}$ to increase marginally to $u_{nt} = 1.05$ in 1961 due to the time lag $\tau = 2$ years from money printing to inflation, with subsequent asymptotic decrease towards $\lim_{t \to T} u_{nt} = 1.00065$ at time $T$. That illustrates a short–term temptation to print money even with equal weights assigned to money printing and inflation.

Figure 1f assumes the time lag $\tau = 2$ years from money printing to inflation and inserts the inflation empirics in Figure 1b into (3) and plots the player's utility $u_{nt}$ for the five Cobb–Douglas elasticities $\alpha = 0.6,\ 0.5,\ 0.4,\ 0.3,\ 0.2$, thus not applying the money printing empirics. Figure 1f also gives overall lower player utility than Figure 1c, possibly because money printing estimated from inflation may cause less estimated money printing than the empirical money printing in Figure 1a. The benchmark elasticity $\alpha = 0.5$, i.e., equal weights assigned to money printing and inflation, causes the player's utility $u_{nt}$ to

increase marginally from $u_{nt} = 1$ in 1959 to $u_{nt} = 1.00995$ in 1960, $u_{nt} = 1.01499$ in 1961, where it remains thereafter.

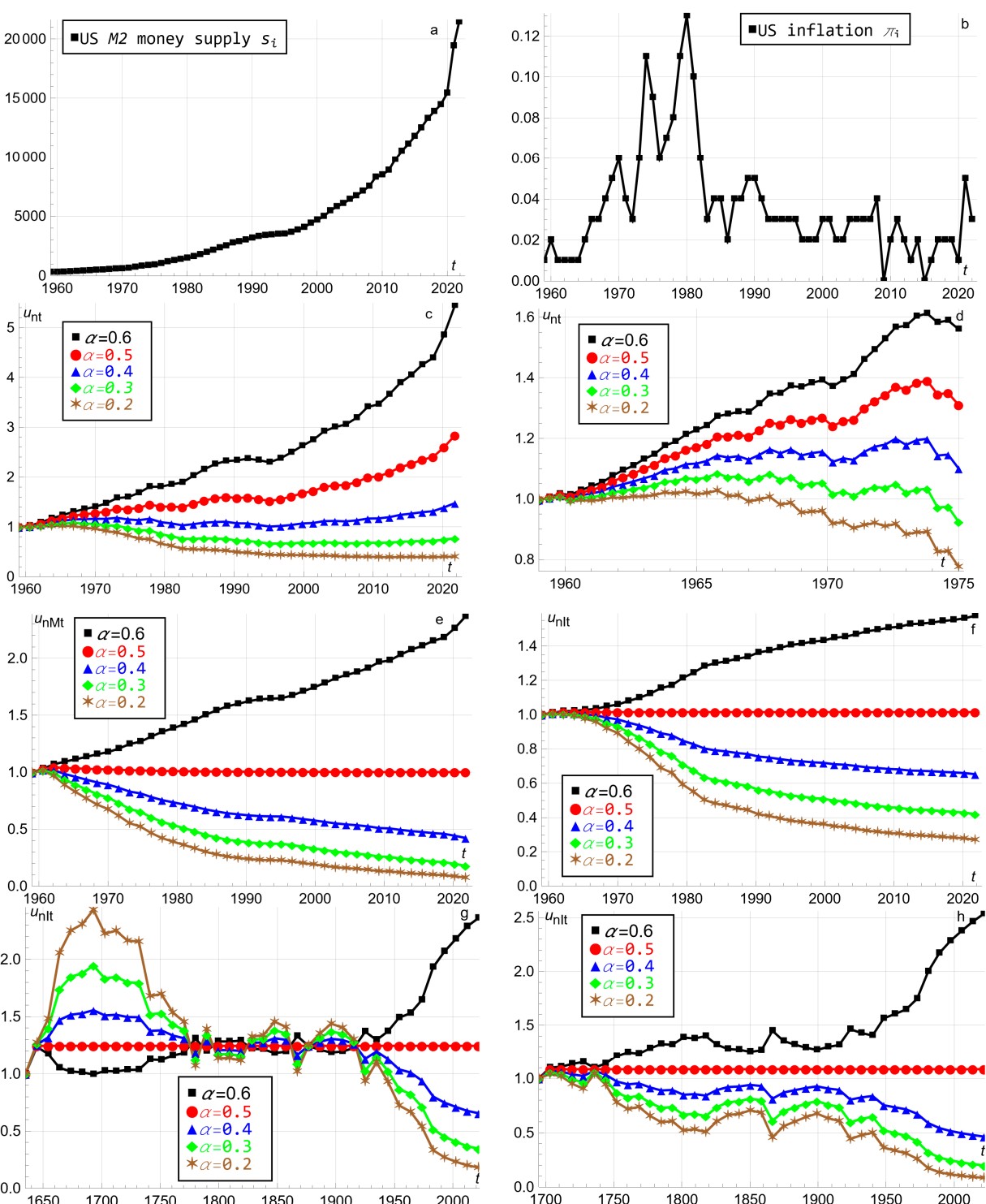

**Figure 1.** Panel (**a**): US M2 money supply $s_i$ 1959–2021 in USD billion. Panel (**b**): US inflation $\pi_i$ 1959–2021. Panels (**c**–**h**): The player's utility $u_{nt}, u_{nMt}, u_{nIt}$ as a function of time $t$ for the Cobb–Douglas elasticities $\alpha = 0.6, 0.5, 0.4, 0.3, 0.2$. Panel (**c**): Equation (1) 1959–2021. Panel (**d**): Equation (1) 1959–1975. Panel (**e**): Equation (2) 1959–2021 based on money printing empirics. Panel (**f**): Equation (3) 1959–2021 based on inflation empirics. Panel (**g**): Equation (3) 1635–2021 based on inflation empirics. Panel (**h**): Equation (3) 1695–2021 based on inflation empirics.

Figure 1g replicates Figure 1f for 1635–2021. High weight $\alpha = 0.2$ assigned to inflation causes the player's utility $u_{nt}$ to be inverse U–shaped and remain above $u_{nt} = 1$ until 1864. That occurs because of the substantial deflation, especially in 1635–1650 (CPI Inflation Calculator 2022). Hence, in contrast, high weight $\alpha = 0.6$ assigned to money printing causes the player's utility $u_{nt}$ to be U–shaped and with minima $u_{nt} = 0.99$ in 1693 and 1695. After 1917, this gets reversed due to less deflation and more consistent inflation. Overall, USD 1 in 2022 buys 2.98% of what it would buy in 1635 (CPI Inflation Calculator 2022).

Figure 1h replicates Figure 1g,f for 1695–2021. The year 1695 is chosen since USD 1 in 2022 buys 1.22% of what it would buy in 1695, which is the lowest percentage for 1635–2021 (CPI Inflation Calculator 2022). Eliminating the 1635–1695 deflation causes Figure 1h to be more reminiscent of Figure 1c–f.

*4.2. Analysis Applying Replicator Dynamics*

Money printing and inflation generally proceed such that the evolution of the fraction $p_t$ of the player's volume of transactions in the variable–supply fiat currency $n$ has no analytical solution.[4] Hence we illustrate the replicator equation in (7) with simulations. Figure 2 applies the same empirics and makes the same assumptions as in Figure 1c, with sensitivity $k = 0.5$, initial condition $p_{nt_0} = 0.5$, and seven different parameters $h_t$ for the player's support of currency $g$ relative to currency $n$ at time $t$.

Figure 2a assumes the Cobb–Douglas elasticity $\alpha = 0.6$, which causes the rapidly increasing player's utility $u_{nt}$ in Figure 1c due to the high weight $\alpha = 0.6$ assigned to money printing. With negligible support $h_t = 0.01$ for the fixed–supply currency $g$, the fraction $p_{nt}$ of the player's volume of transactions in currency $n$ at time $t$ increases rapidly and asymptotically towards $\lim_{t \to 2021} p_{nt} \approx 1$ determined numerically. With increasing support $h_t = 0.3$, $h_t = 0.4$, $h_t = 0.5$ for currency $g$, the fraction $p_{nt}$ increases more slowly towards $\lim_{t \to T} p_{nt} \approx 1$. When $h_t = 0.6$, which means more support for currency $g$ than for the variable–supply currency $n$ at time $t$, the fraction $p_{nt}$ first decreases towards a minimum $p_{nt} = 0.33$ in 1972 since the player's utility $u_{nt}$ in Figure 1c is still too low, and thereafter increases towards $\lim_{t \to 2021} p_{nt} \approx 1$ as the player's utility $u_{nt}$ in Figure 1c increases. When $h_t = 0.7$, the same, but more pronounced logic applies. The difference is that $p_{nt}$ fails to approach $\lim_{t \to T} p_{nt} \approx 1$ approximatively by 2021, but can be expected to do so beyond 2021. Finally, with overwhelming support $h_t = 0.99$ for currency $g$, the high player's utility $u_{nt}$ in Figure 1c is too low when multiplied with $1 - h_t$ in (4). Hence the fraction $p_{nt}$ of the player's volume of transactions in currency $n$ at time $t$ decreases rapidly and asymptotically towards $\lim_{t \to 2021} p_{nt} \approx 0$ determined numerically.

Figure 2b assumes the lower Cobb–Douglas elasticity $\alpha = 0.2$, which initially causes an inverse U–shaped, and thereafter overall decreasing, player utility $u_{nt}$ in Figure 1c due to the low weight $\alpha = 0.2$ assigned to money printing. With low support $h_t = 0.01$ and $h_t = 0.3$ for currency $g$, the fraction $p_{nt}$ increases asymptotically towards $\lim_{t \to T} p_{nt} \approx 1$, but more slowly than in Figure 2a. With higher support $h_t = 0.4$ for the fixed–supply currency $g$, the fraction $p_{nt}$ first increases towards a maximum $p_{nt} = 0.82$ in 1980 since the player's utility $u_{nt}$ in Figure 1c is still too high and thereafter decreases, causing the majority of the volume of transactions in currency $g$, not quite reaching $\lim_{t \to T} p_{nt} \approx 0$ by 2021, but can be expected to do so beyond 2021. With equal support $h_t = 0.5$ for both currencies, the fraction $p_{nt}$ first increases marginally towards a maximum $p_{nt} = 0.505$ in 1967, and thereafter decreases towards $\lim_{t \to T} p_{nt} \approx 0$ with all transactions in currency $g$. With higher support $h_t = 0.5$ for both currencies, $h_t = 0.6$, $h_t = 0.7$, $h_t = 0.99$ for currency $g$, the fraction $p_{nt}$ decreases more quickly towards $\lim_{t \to T} p_{nt} \approx 0$.

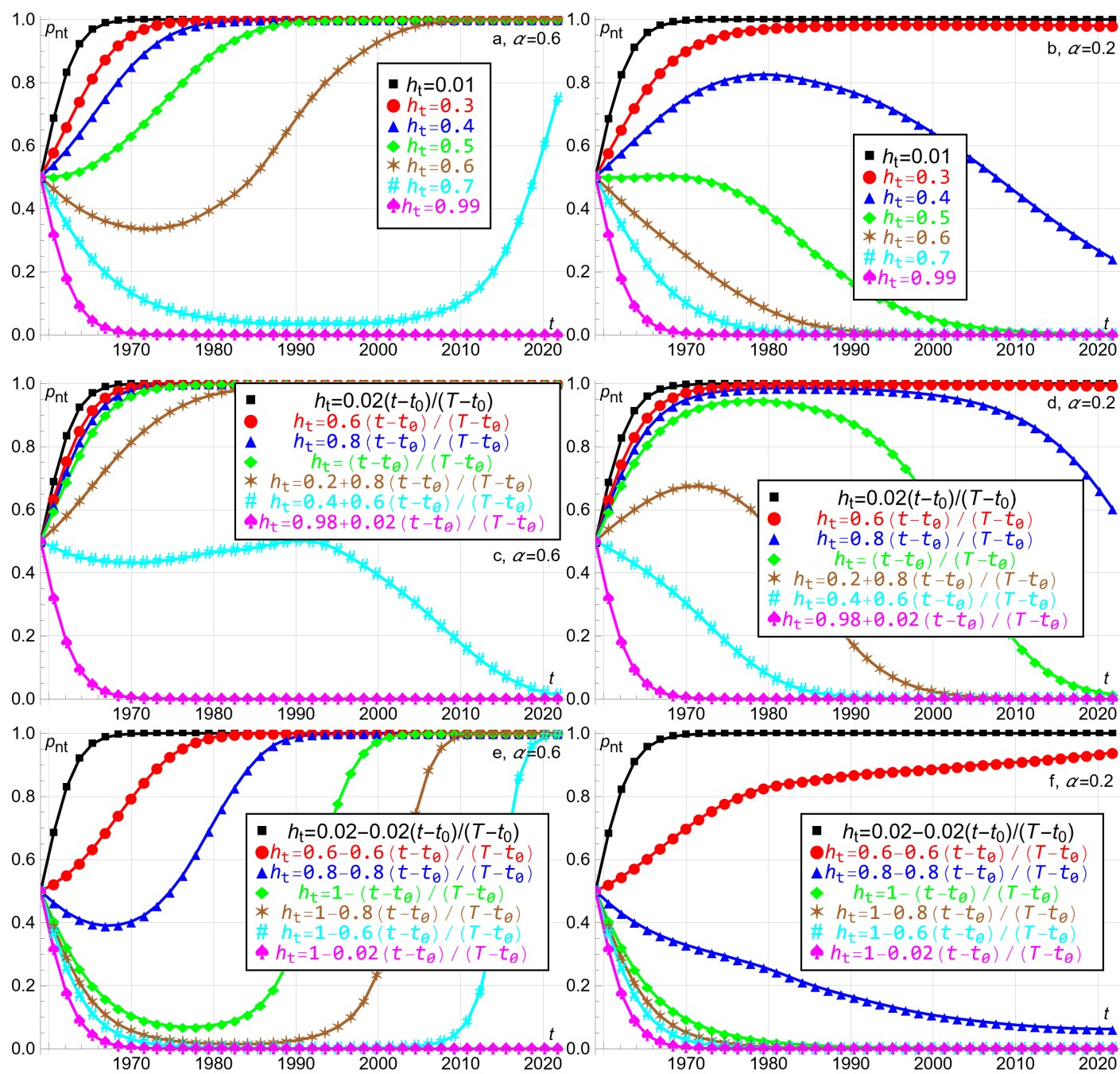

**Figure 2.** The fraction $p_{nt}$ of the player's volume of transactions in currency $n$ at time $t$ 1959–2021 when $k = p_{nt_0} = 0.5$, applying the empirics in Figure 1c. Panels (**a**,**c**,**e**): $\alpha = 0.6$. Panels (**b**,**d**,**f**): $\alpha = 0.2$. Panels (**a**,**b**): Seven constant support parameters between $h_t = 0.01$ and $h_t = 0.99$. Panels (**c**,**d**): Seven linearly increasing support parameters $h_t$. Panels (**e**,**f**): Seven linearly decreasing support parameters $h_t$.

Figure 2c,d assume linearly increasing support $h_t$ for currency $g$, adjusted to equal the support $h_t$ in Figure 2a,b at the midway point $t = t_0 + (T - t_0)/2 \approx 1990$, constrained to be not less than $h_t = 0$ at the initial time $t = t_0$, and constrained to be maximally $h_t = 1$ at the final time $t = T$. Low initial support $h_t$ for currency $g$ means high initial support $1 - h_t$ for the variable–supply currency $c$. Hence the low initial support $h_t$ for the five first linear equations in Figure 2c for $\alpha = 0.6$ causes a more rapid increase in the fraction $p_{nt}$ towards $\lim_{t \to T} p_{nt} \approx 1$ than in Figure 2a. When $h_t = 0.4 + 0.6(t - t_0)/(T - t_0)$, which gives $h_t = 0.7$

at the midway point $t \approx 1990$, the remarkable situation arises where the fraction $p_{nt}$ is initially U–shaped towards the maximum $p_{nt} = 0.5007$ in 1990, and thereafter decreases reaching $\lim_{t \to T} p_{nt} \approx 0.01$ in 2021. This result is the opposite of the result in Figure 2a and arises since the linearly increasing support $h_t$ exceeds $h_t = 0.7$ after 1990, which means more support for the fixed–supply currency g. Hence, although currency $c$ before 1990 enjoys more support in Figure 2c than in Figure 2a, after 1990, the reverse is the case. For the final curve, the results are similar except that the fraction $p_{nt}$ initially decreases more slowly towards $\lim_{t \to T} p_{nt} \approx 0$ than in Figure 2a.

Figure 2d, with the lower Cobb–Douglas elasticity $\alpha = 0.2$, causes more slow asymptotic increase in the fraction $p_{nt}$ towards $\lim_{t \to T} p_{nt} \approx 1$ for the first two linear equations compared with $\alpha = 0.6$ in Figure 2c. Already for the third linear equation with support $h_t = 0.8(t - t_0)/(T - t_0)$, which gives $h_t = 04$ at the midway point $t \approx 1990$, asymptotic increase towards $\lim_{t \to T} p_{nt} \approx 1$ cannot be sustained because of the low weight $\alpha = 0.2$ assigned to money printing. After a maximum $p_{nt} = 0.985$ in 1982, the fraction $p_{nt}$ decreases towards $p_{nt} = 0.60$ in 2021. For the fourth linear equation $h_t$ the maximum $p_{nt} = 0.95$ is reached in 1978, with a subsequent decrease towards $\lim_{t \to T} p_{nt} \approx 0.01$ in 2021. For the fifth linear equation $h_t$ the maximum $p_{nt} = 0.67$ is reached in 1971, with subsequent decrease towards $\lim_{t \to T} p_{nt} \approx 0$ in 2021. For the two final linear equations for $h_t$ the fraction $p_{nt}$ of the player's volume of transactions in currency $n$ at time $t$ decreases relatively rapidly towards $\lim_{t \to T} p_{nt} \approx 0$.

Figure 2e,f assume linearly decreasing support $h_t$ for currency g, adjusted to equal the support $h_t$ in Figure 2a,b at the midway point $t = t_0 + (T - t_0)/2 \approx 1990$, constrained to be maximally $h_t = 1$ at the initial time $t = t_0$, and constrained to be not less than $h_t = 0$ at $t = T$. High initial support $h_t$ for currency g means low initial support $1 - h_t$ for the variable–supply currency $c$. Hence the high initial support $h_t$ for the two first linear equations in Figure 2e for $\alpha = 0.6$ causes more slow increase in the fraction $p_{nt}$ towards $\lim_{t \to T} p_{nt} \approx 1$ than in Figure 2a. For the linear equations number 3, 4, 5, 6 the fraction $p_{nt}$ reaches minima $p_{nt} = 0.39, 0.068, 0.014, 0.0014$ in 1967, 1977, 1983, 1990, respectively, before increasing towards $\lim_{t \to T} p_{nt} \approx 1$ and exhaustive support of the fixed–supply currency $c$. This result arises because the support $h_t$ of the variable–supply currency g is too low and decreasing after 1990. For the final curve, the results are similar except that the fraction $p_{nt}$ initially decreases more rapidly towards $\lim_{t \to T} p_{nt} \approx 0$ than in Figure 2a.

Figure 2f, with the lower Cobb–Douglas elasticity $\alpha = 0.2$, causes a slower asymptotic increase in the fraction $p_{nt}$ towards $\lim_{t \to T} p_{nt} \approx 1$ for the first linear equation compared with $\alpha = 0.6$ in Figure 2e. For the second linear equation the increase is slower. The fraction $p_{nt}$ of the player's volume of transactions in currency $n$ at time $t$ only reaches $\lim_{t \to T} p_{nt} \approx 0.94$ in 2021. Already for the third linear equation an increasing fraction $p_{nt}$ cannot be sustained. Instead, the fraction $p_{nt}$ decreases towards $\lim_{t \to T} p_{nt} \approx 0.06$ in 2021. For the remaining linear equations, the fraction $p_{nt}$ decreases rapidly towards $\lim_{t \to T} p_{nt} \approx 0$ in 2021.

## 5. Summarizing the Results

The article first analyzes the variable–supply currency in isolation. A ratio is established with the initial supply in the denominator and the initial supply plus accumulative money printing (positive) and money withdrawal (negative) in the numerator. A second ratio is established with 1 in the numerator and accumulative inflation (positive) and deflation (negative when measured as a percentage) in the denominator. A Cobb–Douglas utility is established for a player with one output elasticity for each of the two ratios, which are multiplied with each other. The player may be a consumer, firm, organization, or any individual or collective actor conceptualizing a utility for money supply subject to money printing/withdrawal and inflation/deflation. If the output elasticity for the first ratio is

high, money printing/withdrawal is assigned a high weight relative to inflation/deflation, and conversely, if the output elasticity for the second ratio is high. When the two output elasticities are equal, and money printing is outweighed by inflation, or money withdrawal is outweighed by deflation, the product of the two ratios equals 1. When inflation empirics are unavailable, a second utility is developed where inflation is calculated from money printing accounting for a time delay. When money printing empirics are unavailable, a third utility is developed where money printing is calculated from inflation accounting for a time delay.

The article shows how the US M2 money supply has increased exponentially since 1959 and how the US inflation has changed since 1635. These empirical data are used to plot the player's utility since 1959 for five different output elasticities. With high output elasticity for money printing, the player's utility has increased overall exponentially since 1959. With lower output elasticity for money printing, the player's utility increases less and eventually decreases overall when money printing is assigned a low weight, which means that inflation is assigned a high weight. Curves such as these provide policy tools for how to weigh the challenging and partly opposing concerns of money printing and inflation against each other. Similar curves are plotted assuming that inflation and money printing empirics, respectively, are unavailable.

The inflation data since 1635 are used to plot the player's utility for the five output elasticities. The strong deflationary periods 1635–1695 imply high utility for assigning high weight to inflation/deflation and thus low weight to money printing (estimated from inflation). Applying the inflation data since 1695 causes the player's utility to be qualitatively similar to the player's utility since 1959. The reason is that USD 1 in 2022 buys 1.22% of what it would buy in 1695, which is the lowest percentage since 1635.

The article next analyzes one variable–supply fiat currency competing with one fixed–supply currency. The latter is assumed to have a certain support that expresses the utility of transacting in it. That support ranges from 0 to 1 and may change over time. A currency's support depends on its backing, convenience, confidentiality, transaction efficiency, financial stability, and security. The Cobb–Douglas utility of the variable–supply fiat currency is multiplied by 1 minus the support of the fixed–supply currency. A player's utility of transacting in both currencies is a weighted sum of the two utilities, weighted by the volume fraction of transactions in each currency. With this conceptualization, the replicator dynamics can be used to determine how the fraction of a player's volume of transactions in each currency evolves over time. The player continuously changes the fraction to maximize its utility.

We first assume a high weight assigned to money printing. With low support for the fixed–supply currency, the fraction of a player's volume of transactions in the variable–supply currency quickly approaches 1. With higher support of the fixed–supply currency, the fraction may temporarily decrease but will eventually increase, except for very high support for the fixed–supply currency.

We thereafter assume a low weight assigned to money printing. Then very low support for the fixed–supply currency still causes the fraction of a player's volume of transactions in the variable–supply currency to approach 1. With higher support of the fixed–supply currency, the fraction may temporarily increase but will eventually decrease, especially for very high support for the fixed–supply currency, in which case the decrease is rapid.

We next consider linearly increasing support for the fixed–supply currency over time. With high weight assigned to money printing and low but linearly increasing support for the fixed–supply currency, the fraction of a player's volume of transactions in the variable–supply currency approaches 1 quickly. With higher and linearly increasing support for the fixed–supply currency, the fraction may increase temporarily and eventually decrease. Conducting the same analysis with a low weight assigned to money printing may cause the fraction to increase temporarily and thereafter decrease.

We finally analyze linearly decreasing support for the fixed–supply currency over time. With high weight assigned to money printing and low or intermediate, and linearly

decreasing support for the fixed–supply currency, the fraction of a player's volume of transactions in the variable–supply currency may decrease temporarily and thereafter increase towards 1. Conducting the same analysis with a low weight assigned to money printing may cause the fraction to increase for low and decreasing support for the fixed–supply currency and to decrease with slightly higher and decreasing support for the fixed–supply currency.

## 6. Discussion, Policy Implications, Limitations, and Future Research

Research on cryptocurrencies has increased in recent years. Examples of foci are how cryptocurrencies, such as, e.g., Bitcoin compete with fiat currencies such as CBDCs, and the impact of cryptocurrencies on monetary policy, fiscal policy, welfare, and disintermediation of commercial banks. In this context, this article's analysis builds intuition on some aspects of the currency competition between a variable–supply currency and a fixed–supply currency.

First, the article provides insight for policymakers by focusing on two features of competing currencies, i.e., supply and inflation/deflation. A player's support of one currency relative to the other currency is analyzed. A poorly supported currency is prone to decreasing prevalence in the long run. The findings provide useful insights for central banks and governments seeking to adjust the money supply, inflation rate, and the currency's support in the presence of multiple currencies.

Second, the replicator equation presents the evolution of the volume fractions of the two competing currencies. The Cobb–Douglas elasticity for money printing, the Cobb–Douglas elasticity for inflation, and the player's support for one currency relative to the other currency determine the player's volume fraction of transactions in each currency evolutionarily. Therefore, in addition to the money supply and inflation/deflation, policy makers may account for the support of a currency when setting monetary policy.

Third, considering the importance of support for a currency by many different actors beyond the one player modeled in this article, central banks may analyze the sources of support for various currencies, e.g., backing, convenience, confidentiality, transaction fees, transaction efficiency, financial stability, security, purchasing power risk, privacy, etc. The central bank may thereafter choose measures to improve the support of its own fiat currency, in daily use, for borrowing and saving, for cross–border payments, etc.

Fourth, financial investors, individuals, and cryptocurrency developers may find it beneficial to understand the backing of the various currencies when making decisions.

Fifth, the findings provide insights for policy analysis based on money printing/withdrawal, inflation/deflation, and currency support, which determine the volume fractions of transactions in the various currencies. The different degrees of money printing/withdrawal, inflation/deflation, and currency support cause various outcomes.

Sixth, in this digitalized era, central banks around the world are embracing CBDCs.[5] At the time of writing, 105 countries, representing over 95 percent of global GDP, explore CBDCs. Eleven countries have already launched CBDCs. CBDCs may face various challenges, perhaps especially from various cryptocurrencies such as Bitcoin. Central banks may enhance CBDCs' competitiveness by implementing policies aimed at improving the backing of CBDCs, reducing transaction frictions, limiting inflation, and improving the payment environment.

Seventh, the results indicate how a player may transform into using one variable–supply currency and one fixed–supply currency or a combination of two currencies through evolutionary dynamics. This, in turn, may affect the financial markets, monetary policy, fiscal policy, taxes, cross–border payments, etc. Therefore, central banks may pay more attention to the independence and effectiveness of the monetary policy and fiscal policy when facing currency competition. The evolution and adoption of a non–fiat cryptocurrency might potentially undermine the effectiveness of the current monetary policy. This article intends to shed light on how this evolution may play out.

Overall, the article provides policy implications on how to weigh the challenges deriving from money printing, inflation/deflation, and the relative support of variable–supply and fixed–supply currencies.

The two currencies case is the simplest case for multiple currencies. This article seeks to capture the essentials of the phenomenon by focusing on the simple case of competition between two currencies, assuming that one currency has a variable supply while the other currency has a fixed supply. Analyzing only two currencies is also a limitation since today's world has more than two currencies. The evolution and potential stationary coexistence of multiple currencies may be explored in future research. To address further limitations, future research may analyze currency competition accounting for characteristics other than supply and inflation and alternatives to the money–in–the–utility function. Different time preferences and risk attitudes may be assessed. Empirics from countries other than the US may be considered. Different kinds of players with different preferences may be incorporated. Governmental regulation and taxation may be included. Other approaches for incorporating multiple currencies may be assessed, e.g., substitution, individual preferences, switching costs, and fractions of prevalence for various currencies.

## 7. Conclusions

This article analyzes variable–supply and fixed–supply currencies and competition between digital currencies. This involves money printing, money withdrawal, inflation, and deflation. Competition between currencies may become more common as digital currencies emerge with different characteristics pertaining to supply, ownership, decentralization, regulation, confirmation of transactions, geographical extension, etc. This article analyzes competition between two currencies focusing explicitly on supply and inflation/deflation. One currency has variable supply, which has been historically the most common. Variable supply means that money can be printed or withdrawn from circulation. Money withdrawal is sometimes referred to as burning money. The other currency has a fixed supply, which means that money can neither be printed nor withdrawn from circulation.

A Cobb–Douglas utility is developed for a player accounting for money printing/withdrawal and inflation/deflation. The article shows how the player weighs these concerns against each other, first for one variable–supply currency in isolation and thereafter in competition with a fixed–supply currency. Empirics are the US M2 money supply 1959–2021 and the US inflation data 1635–2021.

The player's utility is generalized to account for a weighted combination of a variable–supply fiat currency and a fixed–supply currency, accounting for each currency's support which depends on its backing, convenience, confidentiality, transaction efficiency, financial stability, and security. Replicator dynamics illustrate how the player's volume of transactions in each currency evolves over time.

With high weight assigned to money printing, the player eventually prefers the variable–supply currency, which takes longer with moderately higher support of the fixed–supply currency. With low weight assigned to money printing, the same result follows with low support of the fixed–supply currency. However, with higher support for the fixed–supply currency, the player eventually prefers the fixed–supply currency.

With high weight assigned to money printing and low but linearly increasing support for the fixed–supply currency, the player eventually prefers the variable–supply currency. With higher and linearly increasing support for the fixed–supply currency, the player eventually prefers the fixed–supply currency.

With high weight assigned to money printing and low or intermediate, and linearly decreasing support for the fixed–supply currency, the player may temporarily prefer the fixed–supply currency but will eventually prefer the variable–supply currency.

Finally, low weight is assigned to money printing. Then low and decreasing support for the fixed–supply currency may cause the player to eventually prefer the variable–supply

currency, while slightly higher and decreasing support for the fixed–supply currency may cause the player to eventually prefer the fixed–supply currency.

**Author Contributions:** Both authors contributed to all parts of the article. All authors have read and agreed to the published version of the manuscript.

**Funding:** This research received no external funding.

**Institutional Review Board Statement:** Not applicable.

**Informed Consent Statement:** Not applicable.

**Data Availability Statement:** The article contains no associated data. All data generated or analyzed during this study is included in this published article.

**Conflicts of Interest:** The authors declare no conflict of interest.

## Nomenclature

Parameters

| | |
|---|---|
| $n$ | Variable–supply fiat currency |
| $g$ | Fixed–supply currency |
| $t_0$ | Initial time, $t_0 \geq 0$ |
| $T$ | Final time, $T \geq t_0$ |
| $i$ | Time counting variable, $t_0 \leq i \leq T$ |
| $\tau$ | Time lag from money printing to inflation, $\tau \geq 0$ |
| $s_i$ | Supply at discrete time $i$ of the variable–supply fiat currency $n$, $s_i \in \mathbb{R}$ |
| $\pi_i$ | Inflation at time $i$, $\pi_i \in \mathbb{R}$ |
| $\alpha$ | Cobb–Douglas elasticity expressing weight assigned to money printing, $0 \leq \alpha \leq 1$ |
| $h_t$ | The player's support of currency $g$ relative to currency $n$ at time $t$, $0 \leq h_t \leq 1$ |
| $k$ | Parameter for the sensitivity or rapidity of change of the replicator equation, $k > 0$ |

Independent variables

| | |
|---|---|
| $t$ | Time, $t \geq t_0$ |
| $p_{nt}$ | Volume fraction of the player's transactions in currency $n$ at time $t$, $0 \leq p_t \leq 1$ |

Dependent variables

| | |
|---|---|
| $u_{nt}$ | Player's Cobb–Douglas utility of holding currency $n$ at time $t$, $u_{nt} \geq 0$ |
| $u_{nMt}$ | Player's utility of holding currency $n$ at time $t$ based on money printing, $u_{nMt} \geq 0$ |
| $u_{nIt}$ | Player's utility of holding currency $n$ at time $t$ based on inflation, $u_{nIt} \geq 0$ |
| $u_{ngt}$ | Player's utility of holding currency $n$ at time $t$ when currency $g$ is available, $u_{ngt} \geq 0$ |
| $u_{gnt}$ | Player's utility of holding currency $g$ at time $t$ when currency $n$ is available, $u_{gnt} \geq 0$ |
| $u_t$ | Player's utility of holding currencies $n$ and $g$ at time $t$, $u_t \geq 0$ |

## Notes

[1]  https://coinmarketcap.com/, retrieved 11 July 2022.

[2]  In total, 197,576 metric tons have been mined (gold.org 2022), and 3030 metric tons were produced in 2020 (Basov 2022).

[3]  We may operationalize $h_t$ as comprising six factors, i.e., backing (of currency $n$ relative to currency $g$) by actors, systems, or characteristics that users respect and trust; convenience, e.g., few and easily understood operations when purchasing goods and services; confidentiality, striking balances between privacy, availability, accessibility, and discrimination; transaction efficiency, i.e., low cost, fast speed, affordability, and finality in terms of how many confirmations are needed for transactional approval; financial stability, which usually depends on conditions in the given country; and security, see, e.g., Allen et al. (2020) and Kiff et al. (2020) for the security of blockchain–based currencies.

[4]  For the special case that $k(u_{ngt} - u_{gnt}) = Kt^m$ where $K$ and $m$ are parameters, which depend on time $t$ in a special manner and depend on time $t$ when $m = 0$, the solution of (7) is $p_t = \dfrac{1}{1 + \left(\frac{1}{p_{t_0}} - 1\right)e^{-\frac{K}{1+m}(t^{1+m} - t_0^{1+m})}}$, where $\frac{1}{p_{t_0}} - 1 > 0$ when $0 \leq p_{t_0} < 1$,

$\lim\limits_{t \to \infty} e^{-\frac{K}{1+m}(t^{1+m} - t_0^{1+m})} = 0$ causing $\lim\limits_{t \to \infty} p_t = 1$ when $\frac{K}{1+m} > 0$, $\lim\limits_{t \to \infty} e^{-\frac{K}{1+m}(t^{1+m} - t_0^{1+m})} = \infty$ causing $\lim\limits_{t \to \infty} p_t = 0$ when $\frac{K}{1+m} < 0$, and $\lim\limits_{t \to \infty} p_t = p_{t_0}$ when $\frac{K}{1+m} = 0$. Hence, either one currency excludes the other currency, or the fraction $p_t$ equals the initial fraction $p_{t_0}$ at time $t_0$.

[5]  https://www.atlanticcouncil.org/cbdctracker/, retrieved 12 October 2022.

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
