# Peer review of "Competition between Variable–Supply and Fixed–Supply Currencies"

_economies, doi:10.3390/economies10110270_

Round 1

Reviewer 1 Report

The authors aimed to investigate the, Competition Between Variable-Supply and Fixed-Supply Currencies”, for one variable-supply currency in isolation, multiplied by the inverse of the accumulative inflation/deflation. Overall, the manuscript is well written in all sections of the manuscript, followed by sound methods with results from wide study settings. The study is meaningful in the current situation and context. However, I would like to provide the following comments and suggestions being considered before being accepted for publication. 

·        Abstract: Please re-write the abstract by elaborating more and adding the total number of study subjects as well in addition to study design, and the results should be a bit clearer to the reader.

·        Keywords should be added.

·        In the introduction part, I would like to suggest authors add some related literature, what has already been innovated, discussed; what is new discovery of this paper? Also, I suggest a few more sentences on how this paper will add to the literature gap? How is this paper be beneficial in terms of competition between variable-supply and fixed-supply currencies.

·        Method, could authors first describe the outcome variables in detail

·        Results are expected to be a bit more clear to the reader. Could authors re-write the results section of the paper. 

·        In the discussion section, authors should highlight the strengths of the study while limitations should be extended more precisely noting down biases associated with.

Reviewer 2 Report

1- Literature should be Section 2, instead of 1.3

2- Summary of literature need to included in the end of literature which reports the gap, in light of existing literature.

3- Section 1.4 is not needed.

4- On page 4, the models values need to reported in line and its better to be reported after the equations.

5- Resolve issue on page 9. "Error! Reference source not found"

6- Conclusion need to be reduced and report the policy suggestion of the study.

Round 2

Reviewer 2 Report

The manuscript is acceptable in present form